# The Acute and Short-Term Inhalation of Carbon Nanofiber in Sprague-Dawley Rats

**DOI:** 10.3390/biom12101351

**Published:** 2022-09-22

**Authors:** Mi Seong Jo, Boo Wook Kim, Young Hun Kim, Jin Kwon Kim, Hoi Pin Kim, Jae Hoon Shin, Gun Ho Lee, Kangho Ahn, Mary Gulumian, Il Je Yu

**Affiliations:** 1HCT Co. Ltd., Icheon 17383, Korea; 2Institute of Health and Environment, Seoul National University, Seoul 08826, Korea; 3College of Medicine, Chung-Ang University, Seoul 06974, Korea; 4Occupational Lung Diseases Research Institute, KCOMWEL, Incheon 21417, Korea; 5Department of Mechanical Engineering, Hanyang University, Ansan 15586, Korea; 6Haematology and Molecular Medicine, University of the Witwatersrand, Private Bag 3, Johannesburg 2050, South Africa; 7Water Research Group, Unit for Environmental Sciences and Management, North University, Private Bag X6001, Potchefstroom 2520, South Africa

**Keywords:** carbon nanofiber, inhalation toxicity, acute, short-term inhalation study (STIS)

## Abstract

The inhalation toxicity of carbon nanofibers (CNFs) is not clearly known due to relatively few related studies reported. An acute inhalation study and short-term inhalation study (5 days) were therefore conducted using Sprague-Dawley rats. In the acute inhalation study, the rats were grouped and exposed to a fresh air control or to low (0.238 ± 0.197), moderate (1.935 ± 0.159), or high (24.696 ± 6.336 mg/m^3^) CNF concentrations for 6 h and thereafter sacrificed at 14 days. For the short-term inhalation study, the rats were grouped and exposed to a fresh air control or low (0.593 ± 0.019), moderate (2.487 ± 0.213), or high (10.345 ± 0.541 mg/m^3^) CNF concentrations for 6 h/day for 5 days and sacrificed at 1, 3, and 21 days post-exposure. No mortality was observed in the acute inhalation study. Thus, the CNF LC_50_ was higher than 25 mg/m^3^. No significant body or organ weight changes were noted during the 5 days short-term inhalation study or during the post-exposure period. No significant effects of toxicological importance were observed in the hematological, blood biochemical, and coagulation tests. In addition, the bronchoalveolar lavage (BAL) fluid cell differential counts and BAL inflammatory markers showed no CNF-exposure-relevant changes. The histopathological examination also found no CNF-exposure-relevant histopathological lesions. Thus, neither acute nor 5 days inhalation exposure to CNFs induced any noticeable toxicological responses.

## 1. Introduction

The development of nanotechnology has given many opportunities to electronic engineering, material engineering, environmental technology, and life science. In particular, the application of nanotechnology to the biomedical field can bring risks that nanotechnology can bring along with opportunities for new therapeutic and diagnostic technology development. In some cases, biomedical applications are considered due to the superior properties of nanomaterials but are no longer applied due to risk issues [1,2,3]. Carbon nanofibers (CNFs), defined as nanofibers composed of carbon, are widely used in commerce due to their broad range of applications [4]. CNFs are applied in many fields, such as photocatalytic, nanocomposites, energy devices, filtration, sensors, and biomedical applications, including tissue engineering and drug delivery [5]. CNFs are manufactured in two ways: chemical vapor deposition using transition metal catalysts or electrospinning. CNFs can also be generated as impurities during the synthesis of carbon nanotubes (CNTs) [6]. While CNTs are also manufactured using chemical vapor deposition, CNTs and CNFs manifest many differences that could lead to different toxicological responses. CNTs have a greater aspect ratio than CNFs and contain fewer metal catalysts due to their high heat treatment during manufacturing. In addition, CNTs have more Brunauer Emmett Teller (BET) surface area than CNFs.

However, when compared with CNTs, relatively few studies have evaluated the inhalation toxicity of CNFs. In one study, C57BL/c mice were exposed to a suspension of vapor-grown CNFs (120 μg/mouse) based on a single administration of pharyngeal aspiration. The mice were then sacrificed at 1, 7, or 28 days after the aspiration exposure. The exposure resulted in oxidative stress due to the accumulation of 4 hydroxynonenal (4 hNE) and carbonylated proteins in the lung tissues. Moreover, at 28 days post-exposure, local inflammatory and fibrogenic responses were accompanied by a modified systemic immunity with a decreasing proliferation of splenic T cells ex vivo [7]. In another, CNF aspiration exposure experiment, C57BL/c mice were exposed to vapor-grown CNFs (VGCNFs) at 40 μg and 120 μg/mouse via pharyngeal aspiration and sacrificed after one year. As a result, the CNFs were found to induce chronic bronchopneumonia and lymphadenitis, accompanied by pulmonary fibrosis. This one-year post-exposure study revealed an increased incidence of *K-ras* oncogene mutations in the lungs yet no increased lung tumor incidence [8]. The only subchronic (90 days repeated exposure, 6 h/day, for 5 days/week for 13 weeks) inhalation toxicity study of VGCNFs based on OECD test guideline 413 was conducted using male and female Sprague-Dawley (SD) rats. Four groups of rats per sex were exposed nose-only, 6 h/day, for 5 days/week at target concentrations of 0, 0.50, 2.5, or 25 mg/m^3^. Groups exposed to 0 and 25 mg/m^3^ were also evaluated at 3 months post-exposure. The exposed rats showed a small concentration-related detectable accumulation of extrapulmonary fibers with no adverse tissue effects. Moderate and high concentrations also induced an inflammatory reaction in the terminal bronchioles and alveolar duct regions with fiber-laden alveolar macrophage accumulation. In addition, the high-concentration group showed a significant increase in cell damage and inflammation biomarkers such as lactate dehydrogenase (LDH) and microproteins, and inflammatory polymorphonuclear cells (PMN) in the bronchoalveolar lavage fluid (BALF) at 1 day and 90 days post-exposure. Thus, a no-observed adverse effect level (NOAEL) of 0.54 mg/m^3^ was suggested for CNFs for male and female rats based on minimal inflammation in the terminal bronchioles and alveolar duct areas of the lungs after exposure to 2.5 mg/m^3^ [9]. Recently nanofiber-based face masks and respirators as COVID-19 protection have been investigated. Carbon-based air filters, including CNF, are designed to trap air pollutants and fabricated as protective masks due to their high surface area, abundance, stable chemical structure, low resistance, and high functionalization ability with other materials [10]. Filtration efficiency and the release of fibers from nanofiber masks or respirators play significant roles in biomedical applications because these properties can influence respiratory protection and hazard due to their biocompatibility and toxicity, influencing a negative effect on the respiratory system.

Accordingly, the present study investigated the inhalation toxicity of CNFs with a high aspect ratio and large surface area using an acute inhalation toxicity study based on OECD test guideline 403 [11] and a short-term inhalation study (STIS, 5 days) [12] at 1, 3, and 21 days post-exposure. 

## 2. Materials and Methods

### 2.1. Characterization of Carbon Nanofibers

The CNFs (M90, diameter,10~30 nm; length, <10 μm; aspect ratio, 100~; purity wt.%, 85~90) were provided by Carbon Nano-material Technology Co. (Nam-gu, Pohang-si, Gyeongbuk, South Korea) and manufactured as tubular CNFs by aligning graphene layers parallel to the fiber axis. Several CNF manufacturing methods are described in Appendix A. The CNF aerosols were collected on a transmission electron microscope (TEM) grid (Quantifoil 656-200-Cu, Tedpella, Inc., Redding, CA, USA) with a TEM holder (Ecomesure, Saclay, France) and analyzed using a field emission transmission electron microscope (FE-TEM, JEM2100F, JEOL, Tokyo, Japan) equipped with an energy dispersive X-ray analyzer (EDX, TM200, Oxford Instruments plc, Oxfordshire, UK) at an acceleration voltage of 200 kV [13] and Field emission scanning electron microscopy (FE-SEM) (using a polycarbonate filter (PC) filter (size, 27 mm; pore size 0.2 µg, Millipore, Burlington, MA, USA).

### 2.2. Aerosol Generation and Monitoring in Chamber

The specific-pathogen-free (SPF) SD rats were exposed to the CNF aerosols using a nose-only exposure system (HCT, Icheon, South Korea). The CNF aerosols were generated using a generator created by Hanyang University (Ansan, South Korea) (Figure 1), with high efficiency particulate air (HEPA) filter purified air as the carrier gas. Filtered fresh air using a HEPA filter was used for the control groups. The total airflow for each group was maintained at 25 L per minute (L/min) using a mass flow controller (MFX, FX-7810CD-4V, AERA, Tokyo, Japan), where the flow rate to each nose port was 1L/min. The detailed nose-only inhalation chamber system is presented in Appendix A [14]. The target low, moderate, and high concentrations of generated CNFs were 0.25, 2.5, and 25 mg/m^3^ for the acute inhalation study and 0.625, 2.5, and 10 mg/m^3^ for the STIS, respectively. The concentrations of acute toxicity were based on the maximum generated concentration, and the concentrations of STIS were based on 3 mg/m^3^ carbon black threshold limit value (TLV) of ACGIH (American Conference of Governmental Industrial Hygienists) for the moderate concentration. In the acute inhalation study, the mass median aerodynamic diameter (MMAD) was not measured, whereas the particle size distribution was measured using a dust monitor (Model 1.109, Grimm, Ainring, Germany, range 0.25–32 μm). In the STIS, the MMAD was measured using a MOUDI 125NR (Cascade Impactor, MSP Co, Shoreview, MN, USA) with 13 stages (size range; 0.01, 0.018, 0.032, 0.056, 0.10, 0.18, 0.32, 0.56, 1.0, 1.8, 3.2, 5.6, 10 µm) at a flow rate of 10 L/min. Each stage used a polyvinyl chloride (PVC) filter (47 mm, 0.5 μm, SKC Inc., Covington, GA, USA). The aerosol mass collected on the filter was determined by calculating the difference between the post- and pre-weights of the filters. The geometric standard deviation (GSD) of the distribution was derived from the cumulative mass distribution on the filters. The mass concentration of CNFs was determined gravimetrically (as post-weight minus pre-weight) by sampling on a PVC filter (size: 47 mm and pore size 0.5 μm) at a flow rate of 1.5 L/min. 

### 2.3. Animals and Conditions

The 6-week-old (body weight-male; 157.13 ± 0.85, female; 149.94 ± 0.65) SPF SD rats were acclimated for a week. During the acclimation and after inhalation exposure, the animals were housed in polycarbonate cages (maximum of 3 rats per cage) with a controlled temperature (26.51 ± 0.16 °C), and humidity (62.88 ± 1.05%), and 12 h light/dark cycle. The animals were fed a rodent diet (Woojung BSC, Suwon, South Korea) and filtered water ad libitum. During the acclimation period, the animals were trained to adapt to the nose-only inhalation chamber for 6 h/day. For the acute inhalation study, a total of 40 rats were randomly divided into 4 groups: control (*n* = 10), low concentration (*n* = 10), moderate concentration (*n* = 10), and high concentration (*n* = 10) (Table 1). For the STIS, a total of 80 rats were randomly divided into four groups: control (*n* = 20), low-concentration (*n* = 20), moderate-concentration (*n* = 20), and high-concentration (*n* = 20) (Table 1). Since a limited number of animals (20 rats/chamber) can be accommodated in the chamber, 15 male rats were used for post-exposure studies for 1, 3, and 21 days, and 5 female rats were only used for post-exposure 1 day (Table 1). While the control groups received filtered fresh air, the experimental groups were exposed to CNFs for 6 h. for the acute inhalation study and 6 h/day for 5 days for the STIS. The animals were examined daily for any evidence of exposure-related toxic responses. The body weights were measured at the time of purchase, time of grouping, once during the inhalation period, and before necropsy. The food consumption (g/rat/day) was measured once a week. For the acute inhalation study, the rats were observed for 14 days following the 6 h exposure. For the STIS, the rats were allowed to recover for 1, 3, or 21 days following the 5 days of CNF exposure to investigate the inflammatory reaction. The rat experiments were all approved by the Hanyang University Institutional Animal Care and Use Committee in South Korea (HY-IACUC-17-0043).

### 2.4. Organ Weights, Gross Pathology, and Histopathology

The animals fasted from one day before the sacrifice. After collecting blood from the abdominal artery, the animals were killed by Entobar^®^ (Pentobarbiral sodium, 1 mL/kg, Hanlim Pharmaceutical Co, Seoul, South Korea) anesthesia, and the brain, eyes, trachea, heart, thymus, lungs, spleen, kidneys, liver, and testes were carefully removed. These organs were then weighed and fixed in a 10% formalin solution (BBC Biochemical, Washington, DC, USA). The testes were fixed in a Bouin solution, and the eyes fixed in Davison’s solution. The tissues were fixed for 1 week. An automatic tissue tracing device was used (Leica TP1020; Leica Microsystems, Nussloch, Germany) to form paraffin blocks, following treatment with alcohol and xylene. Thereafter, 4–6 µm slices were cut using a microtome device. The slices were then stained with hematoxylin and eosin (BBC Biochemical, Washington, DC, USA), and the slides were examined using light microscopy. For the acute inhalation study, no histopathology was conducted. 

### 2.5. Hematology and Blood Biochemistry

The animals were narcotized with an intraperitoneal injection of Entobar^®^ (1 mL/kg). Blood samples were then collected from the abdominal aorta into Ethylenediaminetetraacetic acid (EDTA) tubes for a hematological assay and a serum separation tube for the blood biochemistry. The blood biochemistry was analyzed using a blood analyzer and blood cell counter (Hitachi 7108, Hitachi, Tokyo, Japan), while the hematology was analyzed using a blood cell counter (Hemavet 0950, CDC Tech., Dayton, OH, USA). The blood coagulation was analyzed using a trisodium citrate tube (Becton Dickinson, Plymouth, UK) and blood coagulation equipment (ACL700, Instrumentation Laboratory, Bedford, MA, USA). For the acute inhalation study, no hematology or blood biochemistry was conducted. 

### 2.6. Bronchoalveolar Lavage (BAL) Cell Analysis and Measurement of Inflammatory Markers in BAL Fluid (BALF)

The BALF was collected from the right lungs, excluding the right caudal lobe. The left lungs were used for histopathology preparation. The right-lung samples were injected with 0.9% NaCl four times (first time with 4 mL and 3 times with 3 mL). The BALFs were then centrifuged for 7 min at 500× *g*, and the BAL cells were collected and re-suspended in 1 mL of 0.9/% NaCl for evaluation [15]. The total cell numbers were determined using a hemocytometer. The BALF from the first sample was also analyzed for inflammatory markers, LDH, micro-total protein (mTP), and micro-albumin (mALB). The BALF biochemistry was analyzed using a blood biochemical analyzer (Hitachi 7108, Hitachi, Tokyo, Japan). The BAL cells were cytocentrifuged and then stained with a Wright Giemsa stain solution to allow a count of the total number of cells, macrophages, and polymorphonuclear cells (PMNs). For the acute inhalation study, the BAL cells and inflammatory markers were not analyzed. 

### 2.7. Lung Deposition Estimation Using MPPD (Multiple-Path Particle Dosimetry) Modeling

The daily lung burden per rat was estimated for 6 h. of continuous exposure, minute ventilation of 0.265 L/min for 5 days exposure [16,17,18] based on their body weight and aerosol properties. The lung deposition fraction was estimated by MPPD (version 3.04, AL, USA). The following calculations were made for the deposited dose (mg/day):Daily deposited dose (mg/day) = average CNF concentration (mg/m^3^) × minute volume (L/min) × exposure duration (h/day) × deposition efficiency (1)

The following calculations were made for cumulative dose (mg/animal)
Cumulative dose (mg)/animal = daily deposited dose (mg/day) × number of day (2)

### 2.8. Statistical Analysis

The statistical analysis was performed using SPSS version 19 (SPSS Inc., Chicago, IL, USA). The data were presented as mean ± standard error (SE). The result evaluations were all performed using an analysis of variance (One-way ANOVA) following multiple comparison tests using Dunnett T and Dunnett T3 (Post hoc analysis). The level of statistical significance was set at *p* < 0.05 and *p* < 0.01.

## 3. Results

### 3.1. Characterization of CNF and CNF Aerosols

The tubular CNFs (TCNFs) were used in the current inhalation study. The physicochemical properties of the CNFs are described in Table 2 and Figure 2. Other metals, including Fe, Mg, Al, Ca, Mn, Cu, Cr, B, Ni, and Zr were detected by Inductively coupled plasma-mass spectrometer (ICP-MS). The surface area analysis of the CNFs measured by the BET Method was 184.41 m^2^g^−1^ (Table 2). The thickness of the mono-fiber CNFs was 32~35 nm and agglomerated CNFs were observed in the FE-SEM and FE-TEM analyses (Figure 3). The CNF aerosols were well maintained in the inhalation chamber during the 6 h/day and 5 days of exposure (Figure 4). For the acute inhalation study, the actual low, moderate, and high mass concentrations were 0.238 ± 0.197, 1.935 ± 0.159, and 24.696 ± 6.336 mg/m^3^, respectively, and for the STIS the actual low, moderate, and high mass concentrations 0.593 ± 0.019, 2.487 ± 0.213, and 10.345 ± 0.541 mg/m^3^, respectively (Table 3). The particle size distributions measured using the dust monitor in the study were 265, 357, and 357 nm for the low, moderate, and high concentrations, respectively. Meanwhile, the MMAD and GSD measured in the STIS were 168 nm (4.24), 280 nm (3.5), and 209 nm (4.3) for the low, moderate, and high concentrations, respectively (Table 3). 

### 3.2. Animal and Conditions

The body weight losses observed in the male and female rats on the 6th day were due to the fast the day before the sacrifice (Figure 5). The animals showed no significant gross effects, body weight losses, or food consumption changes during the acute exposure or 5 days exposure or 21 days post-exposure (Figure 5). As no mortality was observed, the LC_50_ could not be estimated. In the STIS, no significant organ weight changes were noted in the male and female rats (Appendix A), except for a significant increase (*p* < 0.05) of the weight of the testis in the low concentration group when compared with the control group at 21 days post-exposure (Appendix A).

### 3.3. Blood Hematology and Biochemistry

A significant increase (*p* < 0.05) of white blood cells (WBCs) was observed in the male moderate-concentration group compared with the control group at 1 day post-exposure. The mean corpuscular hemoglobin concentration (MCHC) also showed a significant increase in the moderate (*p* < 0.05) and high-concentration groups (*p* < 0.01) at 1 day post-exposure. The absolute neutrophil count was significantly higher (*p* < 0.05) in the control group when compared with the low and moderate-concentration groups at 1 day post-exposure. In addition, the absolute lymphocyte count and absolute large unstained cell count were significantly higher (*p* < 0.05) in the male moderate-concentration group (Appendix A). The mean corpuscular volume (MCV) and mean platelet volume were significantly lower (*p* < 0.05, *p* < 0.01) in the high-concentration group at 3 days post-exposure. Red cell distribution (RDW) was significantly higher (*p* < 0.05) in the male moderate-concentration group when compared with the control group. The percent of lymphocytes was significantly higher (*p* < 0.05) in the male high-concentration group when compared with the control group. The absolute count of lymphocytes was significantly higher (*p* < 0.05) in the high-concentration group when compared with the control group (Appendix A). The percent of unstained basophil cells (BASO) was also significantly higher (*p* < 0.05) in the low-concentration group when compared with the control group at 21 days post-exposure (Appendix A). A significant increase (*p* < 0.05) of WBCs was observed in the female low and high-concentration groups at 1 day post-exposure. In addition, the MCHC showed a significant increase (*p* < 0.05) in the female low and moderate-concentration groups at 1 day post-exposure. The percent of lymphocytes and absolute lymphocyte counts were significantly higher (*p* < 0.05) in the female high-concentration groups when compared with the control group at 1 day post-exposure. At the same time, the MCV was significantly lower (*p* < 0.05) in this female high-concentration group post-exposure (Appendix A).

Inorganic phosphorus (IP) showed a significant decrease (*p* < 0.01) in the moderate and high-concentration groups at 1 day post-exposure. Magnesium (Mg) showed a significant decrease (*p* < 0.01) in the high-concentration group at 1 day post-exposure. Sodium (Na) showed a significant decrease (*p* < 0.05) in the male high-concentration group when compared with the control group at 1 day post-exposure (Appendix A). Creatinine (CRE) showed a significant decrease in the moderate (*p* < 0.05) and high-concentration (*p* < 0.01) groups at 3 days post-exposure. Glutamic-oxaloacetic transaminase (GOT) showed a significant decrease (*p* < 0.05) in the high-concentration group when compared with the control group at 3 days post-exposure. The LDH showed a significant decrease (*p* < 0.05, *p* < 0.01) in the low and high-concentration groups at 3 days post-exposure. IP, creatine kinase (CK), and potassium (K) showed a significant decrease (*p* < 0.01) in the high-concentration group when compared with the control group at 3 days post-exposure. The Na and Mg showed significant decreases (*p* < 0.05, *p* < 0.01) in all the male exposed groups when compared with the control group at 3 days post-exposure (Appendix A). The GOT and LDH showed significant decreases (*p* < 0.01) in the high-concentration group at 21 days post-exposure. IP showed a significant decrease (*p* < 0.05, *p* < 0.01) in all the exposed groups, and Na showed a significant decrease (*p* < 0.05, *p* < 0.01) in the high and moderate-concentration groups when compared with the control group at 21 days post-exposure. The K showed a significant decrease (*p* < 0.05) in the low and high-concentration groups at 21 days post-exposure while Mg showed a significant decrease (*p* < 0.05, *p* < 0.01) in the moderate and high-concentration groups at 21 days post-exposure. The blood urea nitrogen (BUN) showed a significant increase (*p* < 0.05) in the male moderate-concentration group (Appendix A). The IP showed a significant decrease (*p* < 0.05, *p* < 0.01) in the female moderate and high-concentration groups at 1 day post-exposure. The BUN showed a significant decrease (*p* < 0.05) in the female moderate-concentration group at 1 day post-exposure. Triglyceride (TG) showed a significant increase(*p* < 0.05) in the female high-concentration group when compared with the control group at 1 day post-exposure (Appendix A). Taken together, no significant and consistent test substance exposure was related to changes in the blood biochemical and hematological parameters. 

### 3.4. BAL Fluid Studies

The BALF biochemical parameter analysis did not show any significant inflammatory responses in the male and female rats at 1, 3, and 21 days post-exposure (Table 4). The BAL differential cell count, including the macrophages, PMNs, and lymphocytes, were analyzed at 1, 3, and 21 days post-exposure. The total cell count and macrophages were significantly higher (*p* < 0.05) in the male low concentration and significantly lower in the high-concentration groups at 1 day post-exposure. The PMNs were significantly higher (*p* < 0.05) in the high-concentration group when compared with the control group at 3 days post-exposure. The lymphocytes were significantly higher (*p* < 0.05) in the high-concentration group at 21 days post-exposure. None of the female exposed groups showed any significant changes in the BALF biochemical parameters or BAL differential count when compared with the control group (Table 4 and Table 5). These BALF biochemical and lavage cell results indicated that there were no significant, consistent exposure-related inflammatory responses or cell damage during 21 days post-exposure period. 

### 3.5. Blood Coagulation

The blood coagulation parameters were analyzed including the activated partial thromboplastin time (APTT) and prothrombin time (PT). The APTT was significantly higher (*p* < 0.05, *p* < 0.01) in the moderate and high-concentration groups at 1 days post-exposure. The PT was significantly higher (*p* < 0.05, *p* < 0.01) in the moderate and high-concentration groups when compared with the control group at 21 days post-exposure (Table 6). The male exposed groups did not show any significant changes in the APTT or PT at 3 days post-exposure, while the female exposed groups showed no significant changes in the APTT or PT at 1 day post-exposure (Table 6).

### 3.6. Histopathology

The lung histopathology was examined at 1, 3, and 21 days post-exposure for the male rats (Figure 6) and at 1 day post-exposure for the female rats (Figure 7). No inflammatory responses, such as PMN infiltration or alveolar wall thickening were observed in response to the short-term CNF inhalation. A few CNF-laden macrophages were found, yet not on a large scale. 

### 3.7. Estimation of CNF Deposition in the Pulmonary Region

MMAD indicated the generated CNF could be deposited in the pulmonary region. Thus, CNF aerosol deposition in the pulmonary region was estimated by the MPPD. The count median diameter (CMD) of CNF of low (265 nm), moderate (357 nm), and high (357 nm) concentrations were used to estimate the pulmonary deposition. The pulmonary deposition fraction ranged from 0.03–0.067 (Table 7).

## 4. Discussion

The present study evaluated the inhalation toxicity of CNFs using an acute inhalation study based on OECD test guideline 403 and a 5 days short-term inhalation study. Neither study revealed any CNF inhalation toxicity. Therefore, the LC_50_ was higher than 25 mg/m^3^, and the STIS yielded unnoticeable toxicity in the lungs. Also, none of the statistically significant changes, indicating adverse reactions in the hematology, blood biochemistry, and coagulation tests, were noticed in response to CNF exposure. First target organ lung toxicity evaluated by BALF analysis with cell differentiation and inflammatory markers did not show any statistically significant changes after 5 days CNF exposure and 21 days post-exposure period. Furthermore, histopathological evaluation of lung tissue also did not show noticeable changes. 

This study developed a new CNF aerosol generator for generating a desirable aerosol concentration within an MMAD (less than 2 μm) suitable for inhalation toxicity studies. Although OECD acute inhalation toxicity guideline 403 suggests a limit concentration of 2 mg/L (2000 mg/m^3^) [8], this is impossible in the case of CNFs. In the present study, the particle MMAD distribution differences among the concentration groups were due to the dilution of the CNF aerosol using an aerosol dilution system with a single generator. Notwithstanding, the inhalable CNF aerosol MMAD range was less than 2 μm, as suggested by the revised OECD inhalation toxicity guideline [19,20]. The GSD was not within an acceptable range of less than 3. These MMAD and GSD variations were due to the difficulty of generating a uniformly dispersed CNF aerosol from CNF powder that included agglomerates/aggregates with a tangled structure. The generator used in this study was based on a small-scale powder disperser (SSPD) that is compact and suitable for generating aerosols from fibrous materials; however, the limitations include an unstable concentration that is affected by the shape or cohesiveness of the particles, such as CNTs or CNFs. Since the SSPD applies a relatively weak force for dispersing agglomerates, the test substance can remain adhered due to surface adhesion or electrostatic forces [21]. 

The current acute inhalation study of CNFs indicated the following: (1) CNFs cannot be generated on a nanoscale up to the limit concentration (2000 mg/m^3^) suggested by OECD test guideline 403 [11], (2) a coagulation effect influences the monodispersion and polydispersion of particles at the limit concentration, presumably more than 10^7^–10^8^ particle/cc, resulting in a particle size increase [22], (3) a respirable particle MMAD size of less than 2 μm, as suggested by the current revised OECD inhalation test guidelines [12,13], is achievable for CNFs, (4) Globally harmonized system (GHS) hazard classification of CNF is not possible using acute inhalation toxicity testing due to the lack of mortality, and (5) range finding based on acute inhalation toxicity testing is not feasible due to the lack of data. Therefore, an STIS conducted appropriately is an effective alternative to acute inhalation testing and range finding for subacute and subchronic inhalation studies, as an STIS can provide data on aerosol generation stability, concentration ranges for further subacute and subchronic inhalation studies, hazards, or toxicity, as well as toxicokinetic data if the study is conducted appropriately with mass measurement of lung burden [23]. 

When conducting these experiments, there was no consensus quantitative method to measure the lung burden of CNF. Although the deposited amount of CNF in the pulmonary region were estimated without considering clearance during post-exposure, it would have been better to measure actual lung deposition by quantitative method, reflecting clearance kinetics of the deposited CNF. The quantitative and qualitative methods of measuring lung burden after inhalation exposure are now being documented in the international standard (ISO TR/5387) [23]. This Technical Report is published recently and provides information on the measurement of nanomaterial mass in tissue after inhalation exposure, which can inform on lung clearance behaviour and translocation [23]. After lung tissue digestion with acid or alkaline, the collected carbon nanomaterials can be quantified by the elemental carbon analysis (ECA) with High-Performance Liquid Chromatography (HPLC), non-dispersive infrared (NDIR) analysis, Near Infrared Fluorescence Imaging, ultraviolet-visible (UV-Vis) spectrophotometer [20]. Using the method recommended by the standard, the clearance kinetics of nanomaterials in the lung can be measured. 

The TCNFs used in the current inhalation study had similar physicochemical properties to the vapor-grown carbon nanofibers (VGCNF h, Showa Denko KK, Japan) used in a previous 90 days inhalation study based on OECD test guideline 413 and the Pyrograph CNFs used in several aspiration studies [3,4]. While these CNFs are all manufactured using a similar methodology, TCNFs have a thinner diameter, higher aspect ratio, and larger BET surface area when compared with other CNFs. In addition, TCNFs include more impurities, such as Fe_2_O_4_, while VGCNFs have no detectable impurities, and Pyrograf CNFs include a small amount of iron (Table 7). In addition, the lung deposition of TCNFs is much higher when comparing the MMAD between TCNFs (168–987 nm) and VGCNF h (2.0–3.1 μm) (Table 8). 

Yet, despite these physicochemical property differences, the TCNFs in the current study showed similar inhalation toxicity to VGCNF h. Previously, VGCNF h exposure showed minimal infiltration of inflammatory cells in rats exposed to 2.5 mg/m^3^ CNFs, inflammation along with some thickening of the interstitial walls, hypertrophy/hyperplasia of type II epithelial cells, graded as slight for the 2.5 mg/m^3^ concentration, and inflammation of the terminal bronchioles and alveolar duct regions accompanied with fiber-laden macrophage accumulation [9]. However, the current study also showed minimal or non-detectable inflammation with very few CNF-laden macrophages at 10.345 mg/m^3^. These differences may have been due to the duration and exposure concentrations. Nonetheless, according to the nanomaterial or fiber paradigm, TCNFs with a larger surface area and higher aspect ratio with metal impurities would be expected to show higher toxicity when compared to VGCNF h with a smaller surface area and lower aspect ratio with no impurities. This may have been due to the rapid degradation of CNFs when compared with CNTs. CNFs are cylindric nanostructures with graphene layers arranged as stacked cones, cups, or plates, whereas CNTs are CNFs with graphene layers wrapped into perfect cylinders. CNFs have a larger diameter and are less oriented and more disordered, creating different toxicological properties when compared to CNTs that have a smaller diameter and well-oriented ordered structures. Thus, further comparative studies on the biodurability of CNTs and CNFs are needed. Another 90 days subchronic inhalation toxicity of CNFs has been carried out based on the current STIS study with fresh air control and 0.67 (low), 1.27 (moderate), and 2.40 mg/m^3^ (high) concentrations. The results were consistent with the current observations and suggested of 2.4 mg/m^3^ based on no apparent histopathological observations and no concentration-dependent changes of BAL fluid inflammatory markers [24]. 

US NIOSH recommends a recommended exposure limit (REL) of 1 μg/m^3^ elemental carbon as a respirable mass 8 h time-weighted average (TWA) concentration for CNTs and CNFs based on quantitative risk assessment based on animal dose-response data [25]. Many animal data were used to assess the risk of single-wall CNTs (SWCNTs) and multiwalled CNTs (MWCNTs) to propose REL of CNTs. As it was stated earlier, there were no further subchronic or chronic inhalation studies for CNF compared with MWCNTs for a 2-year carcinogenicity study [26]. Tubular CNFs, which have a large surface area and higher aspect ratio expected to show higher toxicity as seen in other nanomaterials, but in this present study, the in vivo inhalation studies did not show higher toxicity. A large portion of metal impurities (~17%) also did not affect toxicity in the present STIS and also in a previous 90 days subchronic study, conducted by the Korea Conformity Laboratories (KCL) [24]. Further studies are needed to identify the toxicity of new varieties of CNFs.

Analyzing a scientific and animal welfare assessment of the OECD test guidelines for the safety testing of chemicals under the European Union REACH System, Combes et al. [27] suggested some limitations in conducting OECD TG 412 that is similar to our setting. They considered that there is a strong case for the experimental design of the study to be critically assessed in relation to the appropriate statistical analysis of the results. A statistical study should be performed to determine whether smaller group sizes could be used.

## Figures and Tables

**Figure 1 biomolecules-12-01351-f001:**
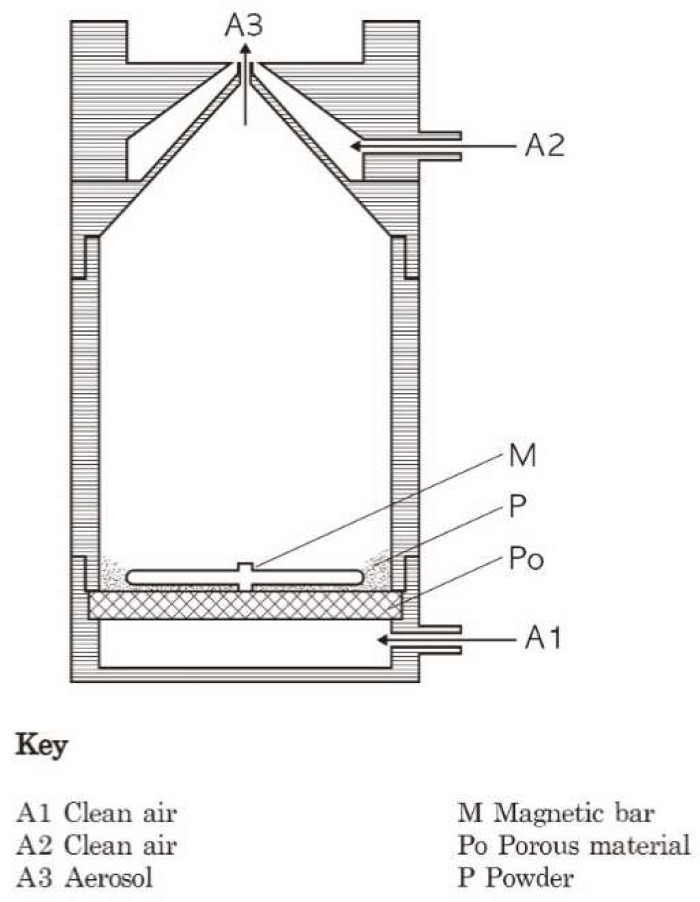
CNF generator. CNF generation scheme. Clean air was introduced into A1 and A2, while CNF powder (P) was stirred using magnetic bar (M). CNF aerosols generated by fluidized bed (A3) were then delivered into inhalation chamber.

**Figure 2 biomolecules-12-01351-f002:**
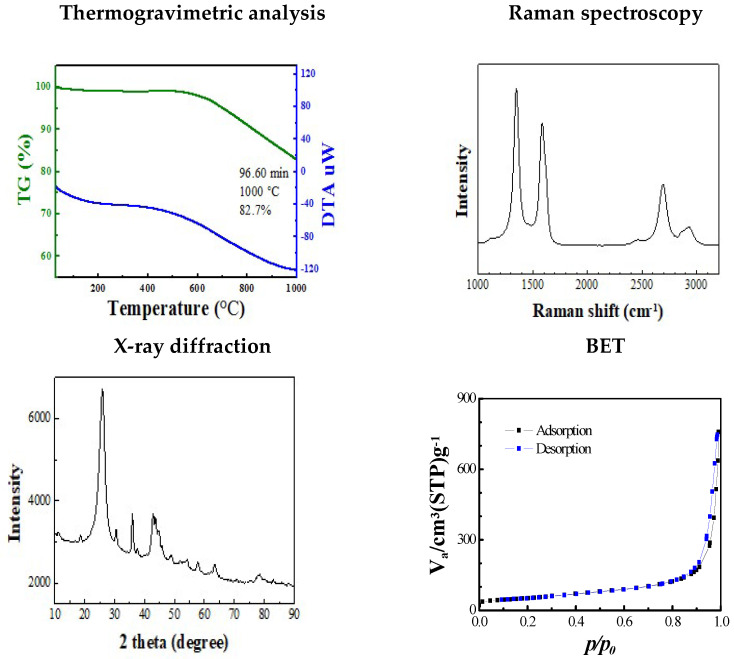
CNF physicochemical characteristics.

**Figure 3 biomolecules-12-01351-f003:**
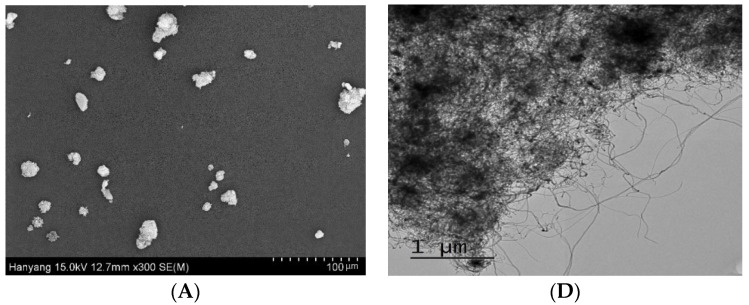
(**A**–**C**) FE-SEM analysis; (**A**) CNF agglomerates (scale: 100 μm); (**B**) high magnification of CNF mono-agglomerate (scale: 200 μm) (**C**) mono-fiber thickness (32~35 nm): (**D**–**F**) FE-TEM analysis; (**D**) agglomerated state of CNFs (scale: 1 μm); (**E**) fibrous state of CNFs (scale: 200 nm); (**F**) high magnification of (**E**) (scale: 50 nm).

**Figure 4 biomolecules-12-01351-f004:**
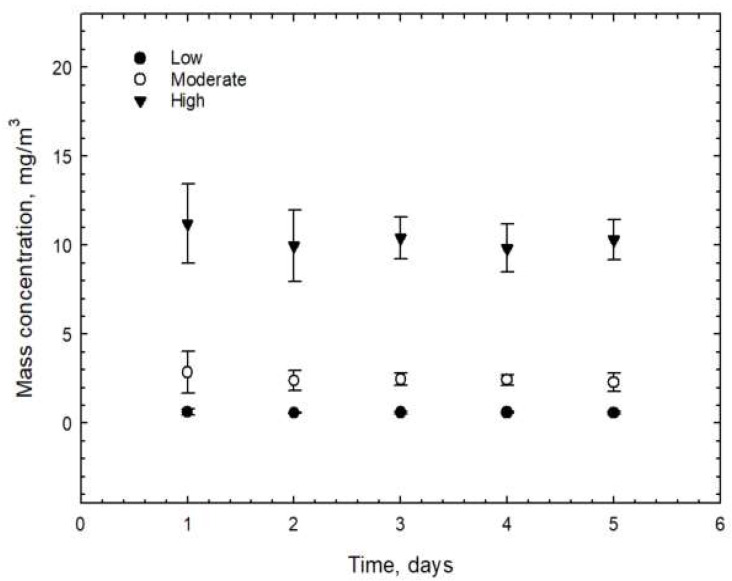
Maintenance of CNF concentrations in inhalation chamber during 5 days exposure.

**Figure 5 biomolecules-12-01351-f005:**
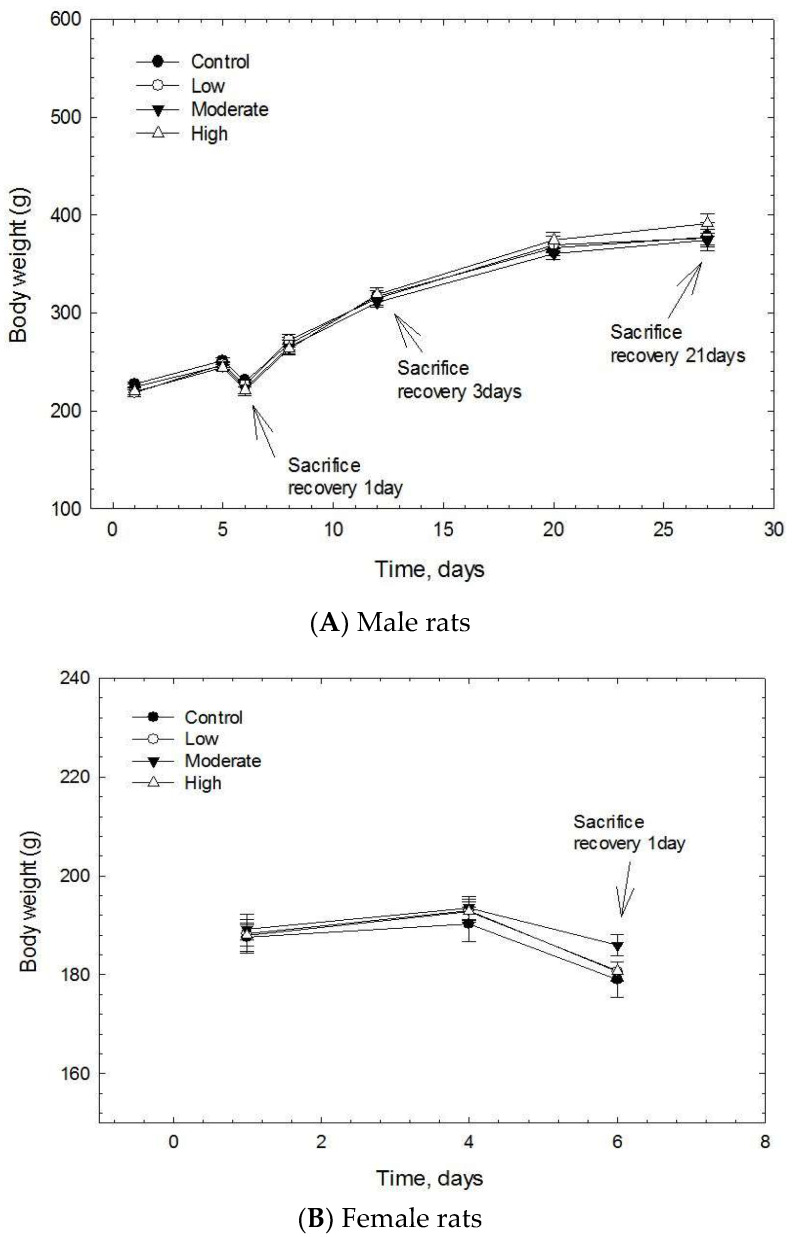
Changes in bodyweights of male rats during 5 days CNF exposure period and at 21 days post-exposure. (**A**), male; (**B**), female.

**Figure 6 biomolecules-12-01351-f006:**
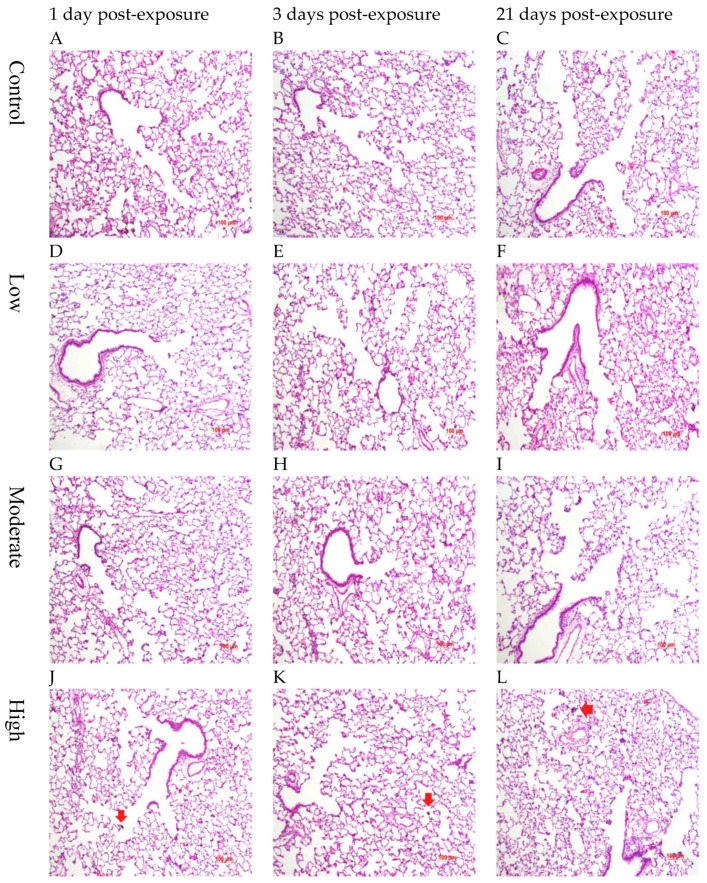
Lung histopathology of male rats at 1, 3, and 21 days after CNF exposure; arrows: alveolar macrophages with ingested CNFs (×100).

**Figure 7 biomolecules-12-01351-f007:**
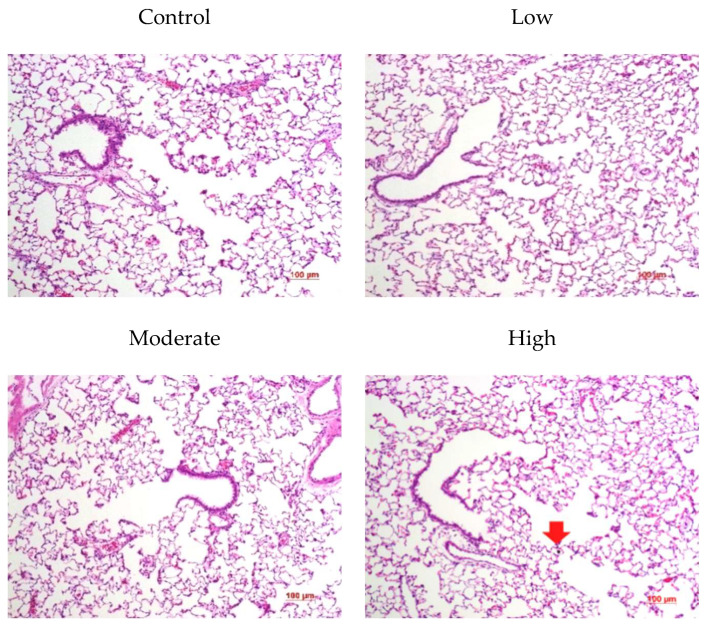
Lung histopathology of female rats at 1 day after CNF exposure; arrow: alveolar macrophage with ingested CNF (×100).

**Table 1 biomolecules-12-01351-t001:** Design of acute inhalation study and STIS.

Acute	STIS		Post-Exposure (PE)		
Concentration	No. of animal	No. of animal	PE-1	PE-3	PE-21
Control	10 (5 M, 5 F)	20 (15 M, 5 F)	5 M, 5 F	5 M	5 M
Low	10 (5 M, 5 F)	20 (15 M, 5 F)	5 M, 5 F	5 M	5 M
Moderate	10 (5 M, 5 F)	20 (15 M, 5 F)	5 M, 5 F	5 M	5 M
High	10 (5 M, 5 F)	20 (15 M, 5 F)	5 M, 5 F	5 M	5 M

M, male; F, female, STIS, short-term inhalation test.

**Table 2 biomolecules-12-01351-t002:** CNF physicochemical information.

Specification	Unit of Measure	Value	Method of Analysis
Carbon content	%	82.7	TGA
Thickness	nm	3~15	TEM
Diameter	nm	10~30	TEM
D/G ratio	-	1.170	Raman
XRD	-	25.8 & 42° peaks	XRD
Purity	Wt.%	85~90	TGA
Surface area	m^2^g^−1^	184.41	BET
Surface charge	mV	−20~−30	ZPA, DLS

XRD, X-ray diffraction; TEM, transmission electron microscopy; ZPA, zeta potential analyzer; BET, Brunauer Emmett Teller method; DLS, dynamic light scattering; TGA, thermogravimetric analysis.

**Table 3 biomolecules-12-01351-t003:** CNF mass concentrations and MMAD in exposure chamber.

Characterization of CNFs in Exposure Chamber			
	Low	Moderate	High
Mass concentration (mg/m^3^)	0.593 ± 0.019	2.487 ± 0.213	10.345 ± 0.541
MMAD (nm)	168	280	209
GSD	4.24	3.75	4.3

MMAD, mass median aerodynamic diameter; GSD, geometric standard deviation.

**Table 4 biomolecules-12-01351-t004:** BALF chemistry for male rats at 1, 3, and 21 days after CNF exposure.

(A) Males (1, 3, and 21 Days Post-Exposure)				
Summary of BALF Chemistry				
				SEX: MALE
Group	Control	Low	Moderate	High
Unit: (mean ± SE)				
1 day post-exposure				
BALF LDH	34.06 ± 2.86 (5)	39.80 ± 4.31 (5)	31.40 ± 2.66 (5)	33.40 ± 4.31 (5)
BALF mALB	10.59 ± 0.41 (5)	9.89 ± 0.52 (5)	13.32 ± 2.38 (5)	15.15 ± 4.10 (5)
BALF mTP	7.60 ± 0.39 (5)	7.76 ± 0.38 (5)	7.91 ± 0.66 (5)	8.88 ± 1.26 (5)
3 days post-exposure				
BALF LDH	33.40 ± 2.79 (5)	31.60 ± 1.94 (5)	33.60 ± 1.63 (5)	39.40 ± 4.77 (5)
BALF mALB	11.33 ± 0.87 (5)	11.53 ± 0.75 (5)	9.82 ± 0.36 (5)	11.73 ± 0.98 (5)
BALF mTP	7.94 ± 0.43 (5)	7.95 ± 0.15 (5)	7.65 ± 0.22 (5)	8.62 ± 0.43 (5)
21 days post-exposure				
BALF LDH	25.00 ± 1.05 (5)	28.00 ± 1.82 (5)	29.60 ± 3.67 (5)	22.80 ± 1.24 (5)
BALF mALB	9.58 ± 0.86 (5)	11.95 ± 2.70 (5)	9.83 ± 1.56 (5)	9.26 ± 1.41 (5)
BALF mTP	7.38 ± 0.18 (5)	8.29 ± 0.79 (5)	7.83 ± 0.54 (5)	7.24 ± 0.38 (5)
**(B) Females (1 day post-exposure)**				
**Summary of BALF chemistry**				
				**SEX: FEMALE**
**Group**	**Control**	**Low**	**Moderate**	**High**
**Unit: (mean ± SE)**				
BALF LDH	28.80 ± 1.77 (5)	32.60 ± 5.27 (5)	32.60 ± 2.20 (5)	44.60 ± 8.24 (5)
BALF mALB	10.74 ± 0.95 (5)	10.70 ± 1.43 (5)	11.99 ± 0.07 (5)	21.69 ± 8.06 (5)
BALF mTP	7.25 ± 0.41 (5)	7.87 ± 0.53 (5)	7.90 ± 0.23 (5)	11.61 ± 3.05 (5)

(): number of animals, LDH, Lactate dehydrogenase; mALB, Micro albumin; mTP, Micro total protein.

**Table 5 biomolecules-12-01351-t005:** BAL differential cell counts at 1, 3, and 21 days after CNF exposure.

(A) Males (1, 3, and 21 Days Post-Exposure)				
Summary of Bronchoalveolar Lavage Analysis				
UNIT: ×10^6^/mL				SEX: MALE
Group: (mean ± SE)	Control	Low	Moderate	High
1 day post-exposure				
Total cell count	4.22 ± 0.23 (5)	7.02 ± 0.34 * (5)	4.68 ± 0.60 (5)	4.05 ± 0.15 * (5)
Macrophages	4.20 ± 0.05 (5)	6.95 ± 0.34 * (5)	4.60 ± 1.33 (5)	3.98 ± 0.33 * (5)
Lymphocytes	0.02 ± 0.01 (5)	0.06 ± 0.01 (5)	0.06 ± 0.03 (5)	0.04 ± 0.02 (5)
PMNs	0.01 ± 0.01 (5)	0.01 ± 0.01 (5)	0.02 ± 0.01 (5)	0.03 ± 0.02 (5)
3 days post-exposure				
Total cell count	5.12 ± 0.32 (5)	6.08 ± 0.45 (5)	5.09 ± 0.16 (5)	4.42 ± 0.31 (5)
Macrophages	5.06 ± 0.71 (5)	5.96 ± 1.00 (5)	4.99 ± 0.35 (5)	4.32 ± 0.67 (5)
Lymphocytes	0.06 ± 0.02 (5)	0.11 ± 0.03 (5)	0.08 ± 0.02 (5)	0.03 ± 0.00 (5)
PMNs	0.00 ± 0.00 (5)	0.01 ± 0.01 (5)	0.03 ± 0.02 (5)	0.07 ± 0.02 * (5)
21 days post-exposure				
Total cell count	8.73 ± 0.81 (5)	7.68 ± 0.28 (5)	8.11 ± 0.13 (5)	6.69 ± 0.20 (5)
Macrophages	8.60 ± 1.77 (5)	7.57 ± 0.64 (5)	7.95 ± 1.96 (5)	6.49 ± 0.46 (5)
Lymphocytes	0.10 ± 0.02 (5)	0.06 ± 0.02 (5)	0.11 ± 0.01 (5)	0.14 ± 0.01 * (5)
PMNs	0.03 ± 0.01 (5)	0.04 ± 0.00 (5)	0.06 ± 0.01 (5)	0.06 ± 0.01 (5)
**(B) Females (1 day post-exposure)**				
**Summary of Bronchoalveolar Lavage Analysis**				
**UNIT: ×10^6^/mL**				**SEX: FEMALE**
**Group:** **(mean ± SE)**	**Control**	**Low**	**Moderate**	**High**
Total cell count	3.44 ± 0.14 (5)	4.24 ± 0.36 (5)	6.11 ± 0.48 (5)	4.82 ± 0.52 (5)
Macrophages	3.39 ± 0.31 (5)	4.20 ± 0.82 (5)	5.96 ± 1.04 (5)	4.72 ± 1.12 (5)
Lymphocytes	0.04 ± 0.02 (5)	0.02 ± 0.01 (5)	0.07 ± 0.03 (5)	0.07 ± 0.04 (5)
PMNs	0.01 ± 0.00 (5)	0.02 ± 0.01 (5)	0.07 ± 0.04 (5)	0.03 ± 0.01 (5)

(): number of animals; PMNs; Polymorphonuclear cells; * *p* < 0.05.

**Table 6 biomolecules-12-01351-t006:** Blood coagulation at 1, 3, and 21 days after CNF exposure.

(A) Males (1-, 3-, and 21 days post-exposure)				
Summary of Blood Coagulation Analysis				
SEX: MALE				
Group	Control	Low	Moderate	High
Unit: (mean ± SE)				
1 day post-exposure				
PT ^1^ (sec)	9.98 ± 0.04 (4)	10.33 ± 0.07 (4)	10.31 ± 0.41 (4)	11.16 ± 0.39 (4)
APTT ^2^ (sec)	13.05 ± 0.35 (4)	14.43 ± 0.73 (4)	16.68 ± 0.59 * (4)	17.60 ± 1.27 ** (4)
3 days post-exposure				
PT ^1^ (sec)	8.73 ± 0.07 (5)	9.09 ± 0.16 (5)	9.12 ± 0.10 (5)	8.82 ± 0.17 (5)
APTT ^2^ (sec)	15.46 ± 1.33 (5)	13.16 ± 0.09 (5)	15.44 ± 0.61 (5)	12.38 ± 0.47 (5)
21 days post-exposure				
PT ^1^ (sec)	8.85 ± 0.16 (5)	9.42 ± 0.31 (5)	9.97 ± 0.21 * (5)	10.18 ± 0.25 ** (5)
APTT ^2^ (sec)	13.38 ± 0.66 (5)	15.66 ± 1.95 (5)	16.22 ± 0.40 (5)	15.28 ± 0.60 (5)
**(B) Females (1 day post-exposure)**				
**Summary of Blood Coagulation Analysis**				
**SEX: FEMALE**				
**Group**	**Control**	**Low**	**Moderate**	**High**
**Unit: (mean ± SE)**				
1 day post-exposure				
PT ^1^ (sec)	9.48 ± 0.20 (5)	9.04 ± 0.26 (4)	10.34 ± 0.90 (5)	9.31 ± 0.25 (5)
APTT ^2^ (sec)	18.36 ± 1.92 (5)	15.35 ± 0.30 (4)	17.82 ± 3.69 (5)	15.38 ± 0.76 (5)

() number of animals; 1. Prothrombin time; 2. Activated partial thromboplastin time; * *p* < 0.05, ** *p* < 0.01.

**Table 7 biomolecules-12-01351-t007:** Deposition of CNF in the pulmonary region during 5 days inhalation exposure.

Concentration	mg/m^3^ (A)	Minute Volume L/min (B)	Daily Exposure Duration (min) (C)	Deposition Fraction in Pulmonary Region (D)	5 days Total Deposition (mg/Lung) (E)
Low	0.593	0.265	360	0.067	0.019
Moderate	2.487	0.265	360	0.03	0.036
High	10.345	0.265	360	0.03	0.148

Daily deposition = A × B × C × D/1000 L; Cumulative dose = D × 5 day; assuming no clearance during 5 days exposure.

**Table 8 biomolecules-12-01351-t008:** Comparison of TCNF, VGCNF h, and Pyrograf CNF.

Name	TCNF (M90)	VGCNF h	Pyrograf CNF
Manufacturing method	CVD	CVD	CVD
Diameter (nm)	16.01 ± 5.78	158	50–160
Length (μm)	<10	5.8	5–30
Aspect ratio	~100	37	-
Purity (%)	85–90	99.5	98.6
Impurity	Fe_3_O_4_	None	0.23% iron
BET surface area m^2^/g	184.41	13.8	35–45
MMAD (μm) (GSD)	0.168–0.280	1.9–3.3 (2.0–3.1)	-

VGCNF h [5], pyrograf CNF [3,4].

## Data Availability

All data are presented in this paper main text and Appendix A.

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
