# Peer review of "The Acute and Short-Term Inhalation of Carbon Nanofiber in Sprague-Dawley Rats"

_biomolecules, 2022, doi:10.3390/biom12101351_

Round 1

Reviewer 1 Report

This study investigated the acute inhalation toxicity of CNFs with a high aspect ratio and surface area on SD rats using a nose-only exposure system. Authors characterized CNF and CNF aerosols and measured many endpoints, including weight change, food consumption, blood hematology, blood biochemistry, BAL biochemistry, blood coagulation and histopathology. Authors failed to justify why they used different numbers of male and female rats in their experiments. The majority of the discussion section talked about their exposure system, general conclusion from the results and comparison to the results from other studies, but failed to discuss their own results.

I. Why did the majority of the difference observed in blood hematology happen in 1-day post-exposure? And why some difference occurred in low concentration groups while other in moderate and high concentration groups?

II. For blood biochemistry, why most endpoints showed significant differences at moderate and high concentration while a few like LDH were significantly affect at low and high concentrations but moderate did not result in significant difference? Similar trend was also observed in BAL fluid studies. Most BAL biochemical parameters were significantly affected at high concentrations while the total cell count and macrophages were significantly affected at low and high concentration but not moderate concentration.

III. Authors measured so many endpoints but failed to systematically analyze these to demonstrate relations between different endpoints and why some differences were only observed in female rats.

IV. The discussion needs major revision.

V. Experimental design (see comment 5 below).

VI. Authors ran hundreds of comparison tests. Did they made adjustments to avoid or reduce the chance of false positive in the statistical results?

Specific comments:

1. μg symbols are in different fonts throughout the manuscript

2. Line 57: delete “(Shvedova et al., 2014)” redundant

3. Line 97-99: Why did authors choose those flow rates in the study?

4. Line 118: Are body weights significant different between male and female rats?

5. Line 127-129: Why did authors used different numbers of male and female rats in their STIS experiment (M:15, F:5)?

    Line 136-137: Why did authors choose all 5 female rats at 1-day exposure and only male in 3 and 21 day exposure.

6. 2.8 Statistical analysis, please indicate if the number used after “±” is standard deviation or standard error.

7. Why type of ANOVA did author use? One-way, two-way or repeated-measured ANOVA?

8. Line 186: add space after 0.238

9. Table 1. For some of the measurements, authors reported range. Are these minimal values and maximum values or 95% confidence intervals or other ranges? Why not use mean ± s.d.?

10. Line 385: “as suggested” is in different font

11. Figure 7, authors only included lung histopathology of female rats at 1 day after exposure due to their experimental design (see comment 5). They had 1, 3 and 21 days for male.

12. Line 429: the manuscript ends abruptly with the comparison between the results of another study and current study.

Author Response

Response to Reviewer 1

This study investigated the acute inhalation toxicity of CNFs with a high aspect ratio and surface area on SD rats using a nose-only exposure system. Authors characterized CNF and CNF aerosols and measured many endpoints, including weight change, food consumption, blood hematology, blood biochemistry, BAL biochemistry, blood coagulation and histopathology. Authors failed to justify why they used different numbers of male and female rats in their experiments. The majority of the discussion section talked about their exposure system, general conclusion from the results and comparison to the results from other studies, but failed to discuss their own results.

Response: Dear Reviewer 1. Thanks for your kind suggestion. We have tried to accommodate all the comments and suggestion in this revision. However, we did not talk about exposure system in the discussion, but we talked about more on test article and aerosol characterization in the discussion that is most important in inhalation studies. Due to the limitation of in vivo CNF inhalation studies, our discussion could be limited. We don’t want to interpret or discuss our study results extensively based on short-term inhalation study.

  1. Why did the majority of the difference observed in blood hematology happen in 1-day post-exposure? And why some difference occurred in low concentration groups while other in moderate and high concentration groups?

Response: Many hematological differences observed in blood chemistry and hematology are not dose dependent-response, which is most important in toxicology. Therefore, we did not regard those responses as test article induced responses. We subsequently placed the tables in supplement information.

  1. For blood biochemistry, why most endpoints showed significant differences at moderate and high concentration while a few like LDH were significantly affect at low and high concentrations but moderate did not result in significant difference? Similar trend was also observed in BAL fluid studies. Most BAL biochemical parameters were significantly affected at high concentrations while the total cell count and macrophages were significantly affected at low and high concentration but not moderate concentration.

Response: Although we described statistically significant differences in blood biochemistry and hematology parameters, there is no significant toxicological response in blood biochemistry and hematology parameters. There was a mistake in the description of the BAL fluid assay; the sentence was corrected. Also, there is no significant exposure related to BAL fluid biochemistry and inflammatory responses.  

III. Authors measured so many endpoints but failed to systematically analyze these to demonstrate relations between different endpoints and why some differences were only observed in female rats.

Response: We presented only important information to the main text and tables. Some data regarded as not important were presented in the supplement. Although the reviewer pointed out “why some differences were only observed in female rats”, We did not see any statistically significant difference in female rats. We believed that the responses were not dose-responses and not important toxicological responses.

  1. The discussion needs major revision.

Response: Discussion was expanded and revised

  1. Experimental design (see comment 5 below).

Response: We inserted Table 1 for the experimental design

  1. Authors ran hundreds of comparison tests. Did they made adjustments to avoid or reduce the chance of false positive in the statistical results?

 Response: We did our best for the statistics.

Specific comments:

  1. μg symbols are in different fonts throughout the manuscript

Response: corrected. If not, the editorial office will do

  1. Line 57: delete “(Shvedova et al., 2014)” redundant

Response: deleted

  1. Line 97-99: Why did authors choose those flow rates in the study?

Response: The flow rate to each port in the nose-only chamber was approximately 1 LPM, similar to the recommended flow of 0.75 L/min by Pauluhn [Pauluhn J, Thiel A. A simple approach to validation of directed-flow noseonly inhalation chambers. J Appl Toxicol. 2007;27:160–7.].

  1. Line 118: Are body weights significant different between male and female rats?

Response: Males are usually heavier than females

  1. Line 127-129: Why did authors used different numbers of male and female rats in their STIS experiment (M:15, F:5)?

Response: The number of exposure ports in the nose-only chamber is limited to 25 ports/chamber. 20 ports were used for animal exposure, and 5 ports were used for test atmosphere monitoring. 

    Line 136-137: Why did authors choose all 5 female rats at 1-day exposure and only male in 3 and 21 day exposure.

Response: Newly added Table 1 shows our experimental design. Since a limited number of animals (20 rats/chamber) can be accommodated in the chamber, 15 male rats were used for post-exposure studies in 1, 3, and 21 days, and 5 female rats were only used for post-exposure 1 day.

  1. 2.8 Statistical analysis, please indicate if the number used after “±” is the standard deviation or standard error.

Response: standard error

  1. Why type of ANOVA did author use? One-way, two-way or repeated-measured ANOVA?

Response: one-way ANOVA

  1. Line 186: add space after 0.238

Response: corrected

  1. Table 1. For some of the measurements, authors reported range. Are these minimal values and maximum values or 95% confidence intervals or other ranges? Why not use mean ± s.d.?

Response: They are range, and these data are obtained from other analytical labs

  1. Line 385: “as suggested” is in different font

Response: corrected

  1. Figure 7, authors only included lung histopathology of female rats at 1 day after exposure due to their experimental design (see comment 5). They had 1, 3 and 21 days for male.

Response: We mentioned the design in Materials and Methods due to the limitation of animal number in the chamber

  1. Line 429: the manuscript ends abruptly with the comparison between the results of another study and current study.

Response: There are a very few in vivo inhalation studies for CNF. We revised the discussion.

Reviewer 2 Report

The manuscript seems interesting but it needs some modifications to be published as a research paper.

1- The language of the manuscript is weak and there are many grammatical mistakes. The manuscript should be revised by a native English speaker.

2- Line 57 of the reference format should be corrected

3- Provide a TEM image of carbon nanofibers

4- The relationship between the structure of the material and the resulting performance is not clearly discussed.

5- There is no discussion in the results and discussion section. In another word, the discussion is not deep enough

Author Response

Response to Reviewer 2

The manuscript seems interesting but it needs some modifications to be published as a research paper.

1- The language of the manuscript is weak and there are many grammatical mistakes. The manuscript should be revised by a native English speaker

Response: Thanks for your suggestion. The manuscript was revised by professor Mary Gulumian who is a native speaker and a native speaker.

2- Line 57 of the reference format should be corrected

Response: corrected

3- Provide a TEM image of carbon nanofibers

Response: TEM images are included in Figure 3 D-F-

‌4- The relationship between the structure of the material and the resulting performance is not clearly discussed.

 Response: The discussion was revised

5- There is no discussion in the results and discussion section. In another word, the discussion is not deep enough

Response: The discussion was expanded and revised

Reviewer 3 Report

Please refer the attached report.

Author Response

Response to Reviewer 3

 General Comments and Recommendations 

This study conducted acute and sub-acute inhalation studies on the potential biological and toxicological effects of carbon nanofibers in specific pathogen-free female and male rats. The authors examined hematological parameters (including coagulation tests), biochemical parameters in bronchoalveolar lavage fluid, and general physical parameters (including organ weights). The study examined effects following a 6-hour acute exposure and following a sub-acute exposure of 6 hours/day for 5 days followed by a 1-day recovery. The sub-acute exposure study also examined these endpoints at 1 (female and male rats), 3 (male rats), and 21 (male rats) days following the 1-day recovery. While there were several statistically significant differences among groups and across recovery periods for several endpoints, no clear patterns were observed (no concentration-response patterns) and the differences are difficult to interpret. To be frank, it is not clear to what extent the rodents in the treatment groups were exposed. While this is an important line of research and this study makes a very limited contribution to the area, it is clear that this research community needs to spend considerably more time on rigor and standards in experimental design and approaches. There are numerous issues and concerns described below that must be addressed. Unfortunately, there are some aspects of this study that cannot be addressed (e. g. robust particle identification, deposition, and quantification in rodent airways). My recommendation is to consider publication only after major revision. 

Response: Unfortunately, we could not determine the content of CNF in rodent airways when we conducted these experiments. There is no standardized method to quantify the deposited CNF in the respiratory tract. Now new methods are emerging, and International standards quantifying carbon nanomaterials are suggested. In this revision, we approached particle deposition by in silico MPPD. We added this particle deposition in the result. 

t

Specific Comments, Questions, and Recommendations 

Introduction 

P2, L51-53: Provide more detail on this cited study specifically the duration of exposure. Was this a one-time instillation? 

Response: a single administration of pharyngeal aspiration as s

P2, L54-55: It would seem that some mice were sacrificed 28 days following exposure so please update L53 accordingly. 

Response: corrected

P2, L57: Please remove the non-numeric citation. 

Response: corrected

P2, L60: Was the exposure period the entire 90-day period, please elaborate? Sub-chronic would seem to indicate that but stating the period again would be helpful. Some readers may not be familiar with toxicology vernacular. 

Response: explained as 6h/day, 5 day/week for 13 weeks

P2: Once you provide an abbreviation for a specific term (LDH for lactate dehydrogenase) you can use that throughout the text going forward except where used in tables or figures and you should spell terms out again in those cases. The authors often repeat terms and their abbreviations throughout the manuscript’s text. 

Response: corrected

Materials and Methods 

P2, L79: Please reconsider your use of punctuation marks. I suggest switching the commas and semicolons. 

Response: corrected

P2, L83: Spell out TEM at first use. Please spell terms out at first use followed by the abbreviation to use going forward throughout the text of the manuscript. I would also say that if the authors use a given term only a few times continue to spell it out otherwise the reader may have to continuously locate the term when first used to understand the abbreviation. 

Response: corrected

P2, L93-94: Why wasn’t the same purified air used as the carrier gas for the CNFs used for the unexposed control mice? 

Response: The purified air was HEPA filtered. We use a separate control HEPA filtered air not to mix with test article air. 

P3, L97-99: What is the justification for these concentrations? Are these concentrations found in settings where humans, presumably workers, are exposed to CNFs? 

Response: The concentrations of acute toxicity were based on the maximum generated concentration, and the concentrations of STIS were based on 3 mg/m3 carbon black threshold limit value (TLV) of ACGIH (American Conference of Governmental Industrial Hygienists) for the moderate concentration.

P3, Figure 1: This figure does not contribute much to the paper or the CNF generator especially B. What are the authors actually trying to visually provide in 1B? 

Response: deleted Figure 1B as you suggested

P4, L127-130: Why is there an imbalance in the samples size for the different sexes in the STIS?

Response: Since a limited number of animals (20 rats/chamber) can be accommodated in the chamber, 15 male rats were used for post-exposure studies in 1, 3, and 21 days, and 5 female rats were only used for post-exposure 1 day. We clarified our experimental design in Table 1.

P4, L131: If the rats in the acute inhalation study were exposed to CNFs for only 6 hours, there is no need to say 6h/day. 

Response: changed to 6h

P4, L137: For the 3 and 21 day STIS rats, these were male only correct? Please specify. 

Response: We provided Table 1. Design of acute and STIS showing animal number and sex

P5 2.8: Why weren’t the data checked for assumptions of normality prior to statistical testing? 

Response: normality was checked by SPSS

Results 

P5 Table 1: There is very little chemical information provided in this analysis. Given the carbon content is a little over 80% that suggests that there are other elements in the remaining fraction that might be important. Can you authors explain this or speculate what other “contaminants” there may be in the CNFs in this study? The XRD suggests other contributors. 

Response: Other metals, including Fe, Mg, Al, Ca, Mn, Cu, Cr, B, Ni, and Zr were detected by ICP-MS (Inductively coupled plasma-mass spectrometer).

P6, Table 2: Can this table be resized to fit within the margins? The authors should footnote the tables and spell out the abbreviations again. Tables and figures should be able to stand alone from the text. 

Response: the tables and figures were revised

P8, Figure 5: This figure appears to show data from both female and male rats, but the figure legend only states male rats. The data also suggest small but significant body mass losses just due to the experimental conditions in the STIS for both female and male rats. The authors need to explain this. 

Response: The body weight loss observed in female rats on the 6th day was due to the feast the day before the sacrifice.

P9, L259: BASO? 

Response: basophil

P11, Table 3 (Females): The group with the highest exposure appears to have notably higher mean values for the biochemical parameters. As is common with toxicology research, variance tends to increase as dose or concentration increases impacting statistical power. This study may not have been sufficiently powered to see statistically significant differences because of the small sample sizes. The authors should consider a post-hoc power analysis and describe what it would take in terms of sample size to achieve statistical significance. 

Response: We used animal numbers suggested by the OECD guideline 412. OECD test guideline for subacute toxicity recommends 10 animals (5 male and 5 female) per concentration group. We believe that there is no significant biochemical parameter change due to the given exposure condition.

General comment: The authors should consider denoting statistically significant differences in all of the relevant tables in the manuscript. They might have considered testing the repeated measures within treatment groups as well to determine if there were any temporal fluctuations or effects. There do appear to be such effects and differences even with the male control group for example in the BALF data. What might explain that if those differences are significant? The use of punctuation marks in the tables especially in the footnoted material is confusing. 

Response If we could use more animal numbers after approval of IACUC, the results could be changed or the same. Since the animal number in our study was based on the OECD 412 (subacute inhalation toxicity), only a limited number of animals per concentration group (5M/5F) was used. Also limitation of exposure chamber ports (20 animals/chamber) also limited our study.

Discussion 

It appears, based on the little data and analysis provided, that very little CNF material made it into the lungs of these rats. It is quite surprising that the authors did not carry out more analysis on the airways, including the upper airways, to determine where CNF material was depositing and if there were any concentration-dependent differences in deposited material. The striking differences in MMAD reported in table 2, which is very inconsistent with what is reported in the preceding text of manuscript, for the STIS study suggest that CNFs behave quite differently at the different concentrations with a concentration-dependent increase in aggregation or agglomeration. This would seem to influence the actual exposures encountered by the rats in the different groups and may explain the highly variable and non-monotonic type responses observed. Rather than just saying there were no clear observable effects, I suggest the authors spend some time recrafting the discussion around the difficulties there study present in terms of assessing and evaluating the toxicology of CNFs and other similar materials.

Response: Thanks for your complete checking of the manuscript. We found an error in the data entry in Table 3 (previous Table 2), and we corrected it. Because our Tubular CNF showed quite different physiochemical properties from vapor-grown CNF (VGCNF) and pyrograph CNF in toxicological outcome 

Round 2

Reviewer 1 Report

The authors have addressed my concerns - the revised manuscript looks great. Recommend for publication. 

Author Response

Thanks for your comments. It helps to improve our manuscript.

Reviewer 3 Report

As requested in the previous round of comments, the authors should conduct a post-hoc power analysis to determine what sample size would have been required to find statistically significant differences. This study was likely underpowered and the justification for sample size selection citing OECD recommendations though well-intended is insufficient to justify this potential experimental design flaw. This results of the power analysis should be included in the manuscript. preferably the discussion where limitations are discussed.

Author Response

Response: we reanalyzed our data again for post-hoc analysis. Also, we mentioned the limitations of experimental design in the discussion.